# Engineered degradation of EYFP-tagged CENH3 via the 26S proteasome pathway in plants

Eberhard Sorge[1]*, Dmitri Demidov[1], Inna Lermontova[1,2], Andreas Houben[1], Udo Conrad[1]*

1 Leibniz Institute for Plant Genetics and Crop Plant Research (IPK), Gatersleben, Germany, 2 Mendel Centre for Plant Genomics and Proteomics, CEITEC, Masaryk University, Brno CZ, Czech Republic

* conradu@ipk-gatersleben.de (UC); Sorge.eberhard@gmail.com (ES)

**Data Availability Statement:** Our original blot/gel image data were posted at a public data repository with the following URL: http://dx.doi.org/10.5447/ipk/2021/1.

## Abstract

Determining the function of proteins remains a key task of modern biology. Classical genetic approaches to knocking out protein function in plants still face limitations, such as the time-consuming nature of generating homozygous transgenic lines or the risk of non-viable loss-of-function phenotypes. We aimed to overcome these limitations by acting downstream of the protein level. Chimeric E3 ligases degrade proteins of interest in mammalian cell lines, *Drosophila melanogaster* embryos, and transgenic tobacco. We successfully recruited the 26S proteasome pathway to directly degrade a protein of interest located in plant nuclei. This success was achieved *via* replacement of the interaction domain of the E3 ligase adaptor protein SPOP (Speckle-type POZ adapter protein) with a specific anti-GFP nanobody (VHHGFP4). For proof of concept, the target protein CENH3 of *A. thaliana* fused to EYFP was subjected to nanobody-guided proteasomal degradation *in planta*. Our results show the potential of the modified E3-ligase adapter protein VHHGFP4-SPOP in this respect. We were able to point out its capability for nucleus-specific protein degradation in plants.

## Introduction

Understanding new aspects of molecular signalling pathways require the elimination of single key proteins within a complex framework. Rapid changes in protein levels *in vivo* allow us to decipher these pathways and address synthetic biology approaches. Trials to eliminate protein-coding genes *via* mutation or CRISPR/Cas9-based methods generally removed the selected protein from all organs, cells and cellular compartments of the organism [1, 2]. Here, a recent study showed an elegant way to reduce protein function in a cell-type-specific manner in plants, based on CRISPR/Cas9 and oestrogen- induceable promotor [3]. However, CRISPR/Cas9 is still error-prone and can show off-target effects [4]. Protein amounts have been lowered *via* the downregulation of the corresponding transcripts using antisense and RNAi strategies [5].

The downregulation of transcript levels reflects only a part of the possible reduction in protein abundance [6]. A selective reduction of proteins could be achieved by degradation *via* the

**Funding:** This research was funded by the German Federal Ministry of Education and Research (Plant 2030, Project 031B0192A, HaploTools). The funders had no role in study design, data collection and analysis, decision to publish, or preparation of the manuscript.

**Competing interests:** The authors have declared that no competing interests exist.

**Abbreviations:** BCR, BTB, CUL3, Ring Box ligase; BTB, Broad complex, tamtrac, bric-à-brac proteins; CENH3, Centromeric histone 3; CUL, Cullin; EYFP, Enhanced yellow fluorescent protein; GFP, Green fluorescent protein; H2B, Histone 2B; MATH, Meprin and TRAF homology domain; POZ, Poxvirus and zinc finger; SCF, SKP1, CUL1, F-box protein; SKP, S-phase kinase-associated protein; SPOP, Speckle-type POZ protein; VHH, Nanobody-derived from camelid single-domain antibody; VHHGFP4, anti-GFP nanobody.

endogenous 26S proteasome, which is a large (~2 MDa) multiprotein complex. Regulated by a cascade of ubiquitin-activating enzyme (E1), ubiquitin-conjugating enzyme (E2) and ubiquitin ligases (E3), single ubiquitin molecules are activated and transferred in an ATP-dependent manner to the lysine ε-amino groups of substrates. At least 4 ubiquitin molecules bound to the target protein are necessary for recognition and depletion by the 26S proteasome [7]. Using a modification of the SCF (Skp1, Cullin3, F-box) E3 ligase, Caussinus and co-workers delivered a proof of concept for the specific protein degradation of GFP-tagged proteins in animal cells [8]. More precisely, they replaced the native WD40 substrate-binding domain of the *Drosophila melanogaster* F-box protein Slmb [9, 10] with a specific anti-GFP nanobody [11] to shift the naturally occurring degradation of E2F [12] to the histone H2B-GFP fusion protein (Fig 1A). Further investigations compared the efficacy of H2B-GFP degradation by four different E3 ligases in mammalian cell lines and zebrafish embryos [13]. The tested VHHGFP-SPOP fusion protein contained the Cullin3 interaction domain BTB, but it was N-terminally fused to the anti-GFP nanobody VHHGFP4, which replaced the naturally occurring MATH domain for target protein interaction [14]. This modified BCR (BTB, Cullin, Rbx1) E3 ligase outperformed any other candidate ligase (including SCF-type ligases) in total nuclear protein degradation, providing promising results for a new spectrum of potential target proteins.

We recently showed that a similar toolkit might be successfully applied in plants with full-length cytosolic EGFP in *Nicotiana tabacum* [15]. Our actual study tested whether chimeric BCR and SCF E3 ligases could be harnessed to degrade nuclear-localised proteins *in planta* (Fig 1A). We showed successful degradation of the target fusion protein EYFP-CENH3 by the chimeric E3 ligase VHHGFP-SPOP in *N. tabacum*. The target protein consisted of an N-terminal enhanced yellow fluorescent protein (EYFP) for detection purposes and a C-terminal centromeric histone H3 variant CENH3 from *Arabidopsis thaliana*. CENH3 is a key component of the centromere, which is required for the proper segregation of chromosomes during cell division and is localised in the nucleosomes of active centromeres [16]. Knockdown of CENH3 in *A. thaliana via* the expression of a CENH3 RNAi construct resulted in a reduced growth rate and fertility [17]. The essential nature of CENH3 was also confirmed by the fact

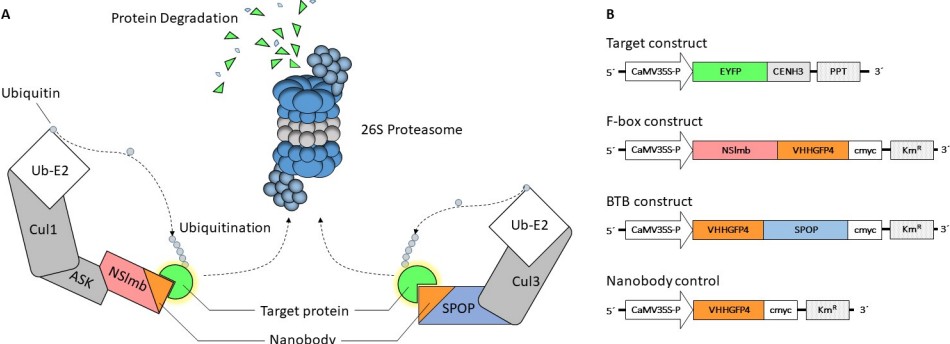

**Fig 1. Principle of ubiquitination by E2-E3 ligase complex and degradation of ubiquitinated proteins.** (A) The E2-E3 ligase complex mediates the transfer of ubiquitin to a target protein. The target specificity depends on the adaptor subunit NSlmb or SPOP. Natural binding domains of SPOP (MATH) or NSlmb (WD40) were replaced with an anti-GFP nanobody, which resulted in chimeric adaptor proteins with high binding affinity for GFP. (B) Schematic presentation of the chimeric proteins used in this study. EYFP-CENH3 located at the centromeres serves as a nucleus-specific substrate. The NSlmb construct originates from the Drosophila melanogaster protein Slmb. The sequence of the SPOP adaptor protein is derived from the human genome. Nanobody control to show the dependency of the directed degradation on the adaptor protein (NSlmb or SPOP). All sequences are under the control of the CaMV35S promoter for plant protein expression. Different selection markers (PPT for phosphinothricin and KM^R for kanamycin resistance) enabled stable double transformant plant lines. C-terminal cmyc tag was used for immunodetection.

that CENH3 mutants were viable only in a heterozygous state [18]. The tagged CENH3 protein localises in centromeric regions of chromosomes within the nucleus and is easy to track *via* fluorescence microscopy. Therefore, CENH3 represents an ideal target protein for directed nucleus-specific degradation.

## Material and methods

### Plasmid construction

The SPOP sequence was provided from the pUC57 standard vector (GeneCust, Boynes, France) according to information from [13]. We performed endonuclease digestion of the pUC57-SPOP vector with *Nco*I and *Bsp120*I (PspOMI) to obtain the insert. The fragment was ligated into the cloning vector pRTRA [19, 20], which was previously cut with *Nco*I and *Not*I. The *Not*I site at the 3' end of the fragment was not recognisable after ligation to the compatible *Bsp120*I site. The resulting pRTRA-SPOP plasmid was also digested with *Nco*I and *Not*I, whose sites were now only located at the 5' end of the SPOP coding sequence. The VHHGFP4 nanobody was cut with the same pair of enzymes and ligated to the linearised pRTRA-SPOP vector, which resulted in pRTRA-VHHGFP4-SPOP. After sequence verification, the expression cassette 35S::VHHGFP4-SPOP was transferred into the binary vector pCB301 *via* compatible *Hind*III sites. Generation of the binary vectors pCB301-Kan-NSlmb-VHHGFP4 and pCB301-Kan-VHHGFP was previously described [15]. Constructs cloned into the pCB301-Kan vector were genetically fused to a C-terminal cmyc sequence (EQKLISEEDLN) and 6x polyhistidine tag. The cloning procedure for the 35S::EYFP-CENH3 vector was described in [16]. Transcription in all constructs was under the control of the CaMV 35S promoter [21, 22].

### Generation of transgenic *Nicotiana tabacum* lines

*Nicotiana tabacum* (SNN) plants were grown in a greenhouse with a 22˚C light phases from 6 am– 10 pm and 20˚C dark phases from 10 pm– 6 am. Leaf discs with a diameter of 0.8 cm were punched out and submerged for 1 h in a transgenic *Agrobacterium tumefaciens* culture containing binary vector with the sequences encoding EYFP-CENH3, VHHGFP4, NSlmb-VHHGFP4 or VHHGFP4- SPOP, plated on Murashige and Skoog medium [23], and stored in an environment controller (Percival Intellus) at 23˚C in the dark for 2 days. Growth conditions were switched to 8-h light followed by 16-h dark. Every 10–14 days, the plant material was placed in fresh MS medium until plantlets appeared. For selection, the plantlets (2–3 cm in height) were placed onto MS medium containing 50 mg/L kanamycin and transferred to a climate-controlled room with a constant temperature of 23˚C and a 16-h light/8-h dark period. High producers of the transgene product were selected using Western blots. Selected plant lines were subsequently transformed with the 35S-EYFP-CENH3 plasmid. Transgenic plants were selected on MS medium containing 50 mg/L kanamycin and 50 mg/L phosphinothricin. Transient expression experiments were done by agroinfiltration [24, 25].

### Immunoblotting

Two leaf discs with an area of 8 mm were punched out from the middle area of an *N. tabacum* leaf and immediately frozen in liquid nitrogen. The material was homogenised using an MM400 CryoMill (Retsch, Haan, Germany) at 28 Hz for 3 min and solubilised in 200 μL 2x SDS sample buffer (100 mM Tris-HCL pH 6.8, 4% SDS, 20% glycerol, 2% -mercaptoethanol, 25 mM EDTA, 0.04% bromophenol blue). After heating at 95˚C for 10 min and centrifugation for 30 min at 21,000 x g, the protein concentration of the total extract was determined using

the Bradford assay (Bio-Rad). For this purpose, 0.5 μl of each extract were mixed with 1 mL Bradford solution. Control values (0.5 μl 2xSDS sample buffer in 1 mL Bradford solution) were subtracted. Total protein extract (10 μg) was loaded onto a 7–15% gradient SDS-PAGE gel [26] with a 5% stacking gel and separated for 2 h at 100 V using a Mini Protean® Tetra Cell system (Bio-Rad). Proteins were transferred to an unmodified 0.45-μm nitrocellulose blotting membrane (GE Healthcare Lifescience) using the wet blot technique. The detection of proteins with a cmyc tag using monoclonal anti-cmyc antibodies was performed as described by [27]. The secondary antibody was a sheep ECL™ anti-mouse IgG horseradish-peroxidase-linked whole antibody (GE). ECL development was used for signal detection in both cases.

Semiquantitative determination of EYFP and AtCENH3 protein amounts in plants was done using Western blotting of nuclear proteins. Isolation of nuclei was performed as described previously [28]. Aliquots (30 μg) of nuclear protein extracts were analysed using 10% Tris-tricine PAGE [26] and stained in Coomassie Blue or electro-transferred onto Immobilon TM PVDF membranes (Millipore, www.merckmillipore.com). The membranes were incubated with anti-GFP (Chromotek, www.chromotek.com), anti-histone H3 (Abcam, www.abcam.com), or a polyclonal anti-CENH3 antibody against *A. thaliana* (LifeTein (Hillsborough, New Jersey US)). Thereafter, they were incubated for 12 h at 4˚C in PBS containing 5% w/v low-fat milk powder to saturate free binding sites. The membranes were incubated in a 1:1,000 dilution of primary antibody (PBS containing 5% w/v low-fat milk powder) for 1 h at 22˚C. Proteins bound by antibodies were detected with 1:5,000 diluted secondary antibodies using Odyssey as recommended by the manufacturer.

### RNA isolation and qPCR analysis

Total RNA was isolated from young leaves using the Qiagen RNeasy plant mini kit (http://www.qiagen.com/).To avoid contamination with genomic DNA, total RNA used for RT-PCR analysis was treated with Turbo DNA free (ThermoFisher Scientific). Reverse transcription was performed using a first-strand cDNA synthesis kit (Thermo Fischer Scientific), oligo (dT) 18 primer and 2 μg of total RNA as the starting material.

Quantitative real-time measurements were performed using POWER SYBR Green Master Mix reagent in a QuantStudio 6 Flex system (Applied Biosystems, Thermo Fisher Scientific, Waltham, MA, USA), according to the manufacturer's instructions. The cDNA equivalent to 40 ng of total RNA was used in a 10 μL PCR. EYFP–CENH3 transcript was amplified using EYFP-specific primers (S1 Table). EF-1alpha-specific primers (S1 Table) were used for control gene amplification. The cycling conditions comprised 10 min polymerase activation at 95˚C and 40 cycles at 95˚C for 3 s and 60˚C for 30 s. Three biological replicates per genotype were tested. Each biological replicate was represented by three technical replicates, which were analysed during the same run. Relative gene expression was calculated using the comparative method $2^{-deldelCT}$ [29].

### Microscopy

Confocal laser scanning microscope LSM 780 (Carl Zeiss GmbH) was used to image leaf sections. Samples were taken from young leaves and mounted onto a microscope slide. EYFP emission was recorded through a bandpass filter with a range of 490–540 nm using a 488-nm laser for excitation. A 20x NA 0.8 objective was used for an overview, and a 40x NA 1.2 water-emersion objective was used for detailed image acquisition. All images were recorded with the same settings to ensure comparability.

## Results

### Preparation of double-transformants co-expressing EYFP-CENH3 and NSlmb-VHHGFP4 or VHHGFP4-SPOP transcripts

The degradation of cytosolic EGFP has been mediated by NSlmb-VHHGFP4 in the cytosol of plant cells [15]. Based on results that showed a nucleus-specific degradation of GFP-fusion proteins by chimeric nanobody-SPOP fusion proteins in mammalian cell lines [13], we investigated whether nucleus-specific degradation is also attainable in plants. In early pilot experiments, we performed transient transformation to test whether the system is capable of degrading stably expressed EYFP-CENH3 by NSlmb-VHHGFP4 or VHHGFP4-SPOP. In transgenic *N. tabacum* lines with stable EYFP-CENH3 expression transiently transformed with a VHHGFP4-SPOP construct, a strong reduction of GFP signals was recognised (S1 Fig). Transient expression of NSlmb-VHHGFP4 in lines with stable EYFP-CENH3 did not lead to degradation but relocalisation of the fluorescence signal.

We generated transgenic *N. tabacum* plants expressing NSlmb-VHHGFP4, which represents an SCF-type E3 ligase, and VHHGFP4-SPOP, which represents a BCR-type E3 ligase, under control of the CaVM35S promoter (Fig 1B). VHHGFP4 without an E3 ligase was used as a negative control. A selected transgenic line was co-transformed with plasmids containing the EYFP-CENH3 fusion construct comprising the *A. thaliana* centromeric histone H3 and enhanced yellow fluorescent protein [16]. The latter construct was also controlled by the CaMV35S promoter.

Twenty double transformants were identified on a selective medium containing phosphinothricin and kanamycin. The presence of the transgene was confirmed by PCR using genomic DNA and primers specific for EYFP, NSlmb-VHHGFP4, VHHGFP4-SPOP and VHHGFP4 only (S2 Fig). PCR products of the expected size (660 bp) for the target construct EYFP-CENH3 and degradation constructs NSlmb (1000 bp), SPOP (1000 bp) and VHHGFP4 (350 bp) were found.

To demonstrate that the transcript level of EYFP-CENH3 did not differ between plants with and without degradation constructs (NSlmb-VHHGFP4 and VHHGFP4-SPOP), quantitative real-time PCR (qRT-PCR) was performed with EYFP specific primers in young leaves of the confirmed lines. Most transgenic plants showed comparable transcript levels, with 0.2 to 1.2 relative expression units (Fig 2). Untransformed wild-type plants served as a negative control. Confocal fluorescence microscopy confirmed expression using the same plant lines (S3 Fig). But Among the 11 tested lines, we observed one outlier with the genotype NSlmb-VHHGFP4/EYFP-CENH3 (line 25). The relative EYFP-CENH3 expression of this line was 12 times higher than that of the other lines; the higher expression was also visible as stronger fluorescence signals in the microscopy images (S3 Fig). Therefore, the presence of a degradation construct does not influence the transcript level of the target gene/protein.

Next, the expression of E3 ligase proteins was determined using Western blot experiments. The chimeric E3 ligases NSlmb-VHHGFP4 and VHHGFP4-SPOP, and VHHGFP4 were detected *via* the CN-terminal cmyc tag. Signals in the expected range (38.4 kDa calculated molecular mass) were obtained for NSlmb-VHHGFP4 (S4 Fig). Lower amounts of the target protein were found in the VHHGFP4-SPOP-expressing lines (37.9 kDa). Control plant lines expressing VHHGFP4 (15.7 kDa) revealed strong signals.

### The target protein EYFP-CENH3 was degraded in VHHGFP4-SPOP/ EYFP-CENH3 tobacco plant lines but not in NSlmb-VHHGFP4/ EYFP-CENH3/ lines

To elucidate whether specific degradation of EYFP-CENH3 by the anti-GFP nanobody-targeted 26S proteasome pathway occurred, transgenic plants were analysed for EYFP-CENH3

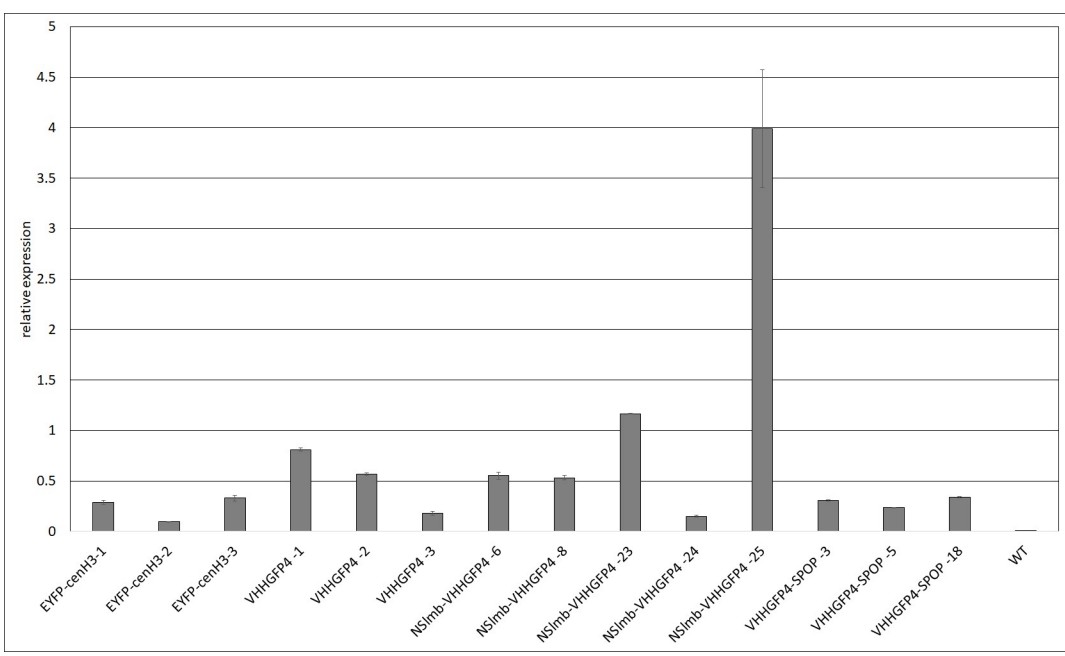

**Fig 2. Quantitative EYFP-CENH3 transcript analysis of transgenic *N. tabacum* lines using qRT-PCR.** Three independent transgenic lines of each genotype were tested for relative EYFP-CENH3 expression. The mRNA levels for EYFP-CENH3 varied between 0.2 and 1.2, except for the NSlmb-VHHGFP4 Line 25 with a value of 4. Fluorescence microscopy analysis confirmed that the high abundance of transcript in line 4 corresponds to the high amount of fusion protein (S3 Fig). Three biological and three technical replicates were analysed for each tested plant line.

using Western blot. We focused on nuclear extracts to analyse the enrichment of the target protein. Anti-*A. thaliana* CENH3 antibodies were used to detect the remaining levels of the EYFP-CENH3 fusion protein in nuclear extracts of transgenic plants (Fig 3A). Samples of EYFP-CENH3-expressing plants exhibited a band at 68 kDa, which might be correlated with the size of a heterodimer that could be formed in centromeric nucleosomes by endogenous *N. tabacum* CENH3 and transgenic EYFP-CENH3 like reported before [30]. The lower accumulation of putative NtCENH3/AtCENH3-YFP heterodimers in the variant with VHHGFP4 compared to NSlmb-VHHGFP4 can be explained by the lower stability of CENH3-EYFP when in complex with VHHGFP4 rather than NSlmb-VHHGFP4. This is partially confirmed by a large amount of CENH3-EYFP monomers in NSlmb-VHHGFP4 transformants.

The Dimerisation of CENH3 has been described not only in plants [25], but also in Drosophila [26, 27]. Besides, it is possible that centromeric heterochromatin of *Nicotiana* does not allow dissociation of CENH3 dimers and separation by SDS polyacrylamide gel electrophoresis even in the presence of a denaturing agent. In plants co-expressing EYFP-CENH3 and VHHGFP4 or NSlmb-VHHGFP4, we observed a 49-kDa band representing EYFP-CENH3. This specific signal was not detected in any other line by anti-CENH3 or anti-YFP Western analysis (Fig 3A, 3B and 3C). All samples, including wild-type, showed an unspecific 41-kDa band. No signals at 49 kDa and/or 68 kDa were found in VHHGFP4-SPOP/EYFP-CENH3 plant lines (Fig 3A and 3B).

Immunodetection of EYFP resulted in similar observations (Fig 3B). We observed the same signal patterns of monomeric EYFP-CENH3 at 49 kDa for VHHGFP4/EYFP-CENH3 and NSlmb-VHHGFP4/EYFP-CENH3 plants. Transgenic plants with VHHGFP4-SPOP/EYFP--CENH3 expression showed no EYFP proteins (Fig 3B). Instead of a loading control via

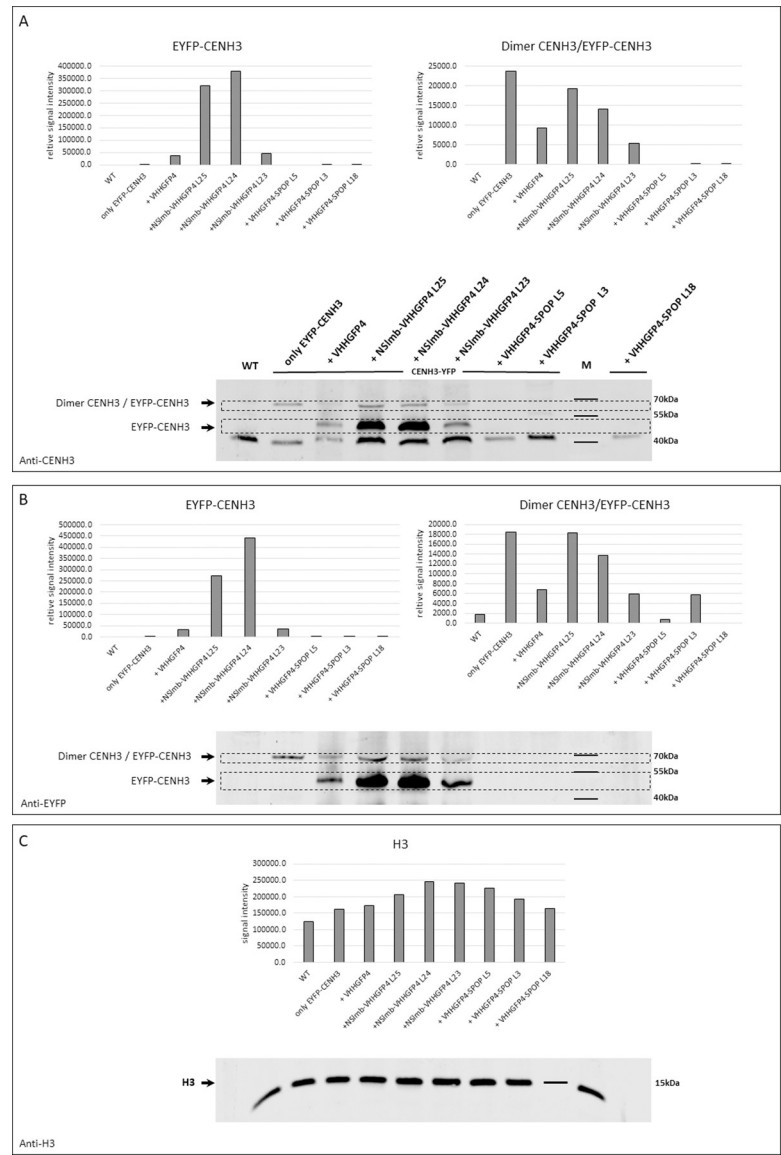

**Fig 3. Target protein abundance in transgenic plants was analysed using Western blot analysis of nuclei extract proteins with anti-CENH3 and anti-EYFP antibodies.** (A) Detection using the anti-*A. thaliana* CENH3 antibody. Control plants expressing EYFP-CENH3 show signals for heterodimers of EYFP-*At*CENH3 and *N.t*CENH3 at 68 kDa. The same results were found for transgenic plants that overexpress VHHGFP4 or NSlmb-VHHGFP4 together with EYFP-CENH3. These lines also exhibited a prominent signal at the range of 49 kDa, which represented EYFP-*At*CENH3. The combination of VHHGFP4-SPOP and EYFP-CENH3 did not show either of these signals. All tested plants, including WT, resulted in an unspecific signal at 41 kDa. (B) Anti-EYFP detection. Control plants with EYFP-CENH3 expression show the same signals described in (a) except for an unspecific band at 41 kDa. This result confirms the signal reduction monitored by anti-CENH3 detection. (C) Anti-histone H3 detection as a loading control. An even distribution of signal intensity represents similar amounts of protein in the assay. Computational quantification of signal strength of single bands was done by "LI-COR Image Studio" software (LI-COR Biosciences–GmbH, www.licor.com) designed for the analysis of Western blot images.

Coomassie staining, we used histone H3 antibodies for Western blotting to demonstrate equal amounts of nuclear protein extracts used for this assay (Fig 3C). We conclude that the target protein EYFP-CENH3 underwent degradation in VHHGFP4-SPOP/EYFP-CENH3 plant lines, but not in NSlmb-VHHGFP4/EYFP-CENH3 lines.

## VHHGFP4-SPOP causes degradation of nuclear-localised EYFP-CENH3 and NSlmb-VHHGFP4 causes mislocalisation of EYFP-CENH3

Confocal microscopy was used to visualise EYFP-CENH3 signals in transgenic plants. Characteristic centromeric dot-like signals were found in EYFP-CENH3 plant lines (Fig 4A and 4B). Similar fluorescent signals were observed in VHHGFP4/EYFP-CENH3 lines (Fig 4C and 4D). The coexpression of NSlmb-VHHGFP4 and EYFP-CENH3 led to a strong accumulation of fluorescent signals in the nucleoplasm, additionally to reduced signals at the centromeres (Fig 4E and 4F). The signal pattern changed from distinct fluorescent dots to diffuse signals, which likely represented non-centromeric chromatin. In contrast, VHHGFP4-SPOP/EYFP-CENH3 plants showed an overall reduction of fluorescence (Fig 4G and 4H). Weak to no EYFP signals were detected in 10 independently tested VHHGFP4-SPOP/EYFP-CENH3/ plant lines (S3 Fig). Residual signals were caused by the autofluorescence of chlorophyll. The weak EYFP-CENH3 signals are consistent with the obtained Western blot results. Together with the transcript data, our analysis confirmed the degradation of the substrate protein EYFP-CENH3. We conclude that VHHGFP4-SPOP is suitable for the degradation of nuclear-localised target proteins, and NSlmb-VHHGFP4 leads to the mislocalisation of EYFP-CENH3. The 26S proteasome-based dependency of target protein degradation was previously demonstrated by the application of the proteasome inhibitor MG132 [15].

## Discussion

The directed degradation of target proteins allows functional studies of selected proteins. We degraded an EYFP-tagged protein that was functionally active in the nucleus. We selected an EYFP-CENH3 fusion protein and co-expressed this fusion protein with two different types of E3 ligases, NSlmb or SPOP, which were fused to an anti-GFP nanobody to generate stably transformed *N. tabacum* plants. The nanobody against EGFP fully cross-reacts with EYFP [31]. The presence of EYFP-CENH3 and VHHGFP4-SPOP was analysed by Western blot analysis. To rule out a partial degradation of the target protein we analysed N-terminal-localised EYFP *via* anti-EGFP antibody and the C-terminal-localised CENH3 *via* anti- CENH3

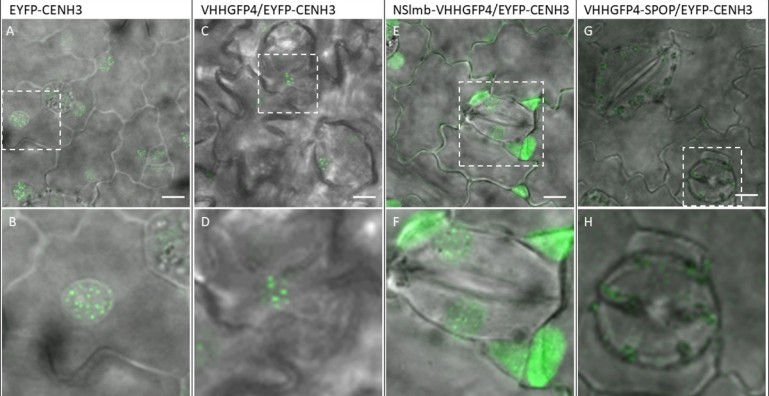

**Fig 4. Confocal microscopy analysis of transgenic *N. tabacum* leaves.** (A and B) Transgenic plants accumulate EYFP-CENH3- within centromeric regions. (C and D) EYFP-CENH3- and VHHGFP4 co-expression. Nanobody without fusion partner binds EYFP but does not result in a signal reduction. Any signal modulation (reduction or translocation) must be induced by the fusion partner (SPOP/NSlmb). (E and F) Coexpression of YFP-CENH3- and NSlmb-VHHGFP4. We observed translocation of the fluorescence signal to the nucleoplasm. (G and H) Co-expression of EYFP-CENH3 and VHHGFP4-SPOP. Strong reduction of the target protein. Scale bar: 10 μm. Magnification 40x. For all genotypes leaf material of 10 independent transgenic lines was analysed.

antibody. Here, co-expression of VHHGFP4-SPOP and EYFP-CENH3 resulted in a significant reduction of the EYFP-CENH3-specific protein signal compared to that in the control lines. In contrast, co-expression of NSlmb-VHHGFP4 and EYFP-CENH3 resulted in the stable and unaltered expression of EYFP protein (Figs 3B and 4E, 4F). The co-expression of the nanobody alone together with EYFP-CENH3 as control also showed stable EYFP protein expression. DNA and mRNA analyses confirmed the presence of EYFP-CENH3 genes and transcripts as prerequisites for expression of this target protein in both combinations (VHHGFP4-SPOP/ EYFP-CENH3 and NSlmb-VHHGFP4/EYFP-CENH3). Western blot results were supported by the epifluorescence data. Neither the co-expression of an anti-GFP nanobody nor the co-expression of NSlmb-VHHGFP4 eliminated the nuclear EYFP signals. Western blot and epi-fluorescence data showed an even higher accumulation of the target protein after co-expression of VHHGFP4 or NSlmb-VHHGFP4. We propose that the nanobody we used for our studies stabilised the target protein, because GFP protein conformations can be manipulated with nanobodies even in living cells [11, 32]. In conclusion, we provide the first evidence for nucleus-specific degradation of proteins in plant cells with the use of the human SPOP E3 ligase. We used only one exemplary protein. The application of this approach to other nuclear proteins is a future task. To explain the presented results in more detail, we must consider that EYFP-CENH3 is translated in the cytosol and likely degraded in the cytosol after co-expression of NSlmb-VHHGFP4 or VHHGFP4-SPOP. Alternatively, the whole complex has been trans-ported into the nucleus for degradation. However, the complex of the E3 ligase and EYFP--CENH3 is too large (>100 kDa) for passive transport *via* nuclear pores in both cases. The SPOP E3 ligases from humans and SPOPs from *A. thaliana* show a tendency to form func-tional dimers [33, 34]. Our studies used the same SPOP sequence as Ju Shin and co-workers [13]. The trimmed version of the SPOP used in our study (no nuclear localisation sequence) must have the ability to form functional dimers with endogenous SPOP as a requirement for successful degradation by the 26S proteasome. Because we noted protein level reduction with this construct, we conclude that human SPOP forms functional dimers with plant SPOP vari-ants. Six different varieties of SPOPs containing a BTB domain were experimentally confirmed at the protein level in *A. thaliana* [33]. Therefore, the presence of endogenous SPOPs in tobacco cells should be expected. The only known functionally interaction partner of this E3 ligase is Cullin3. Two Cullin3 variants in *A. thaliana* were experimentally confirmed at the protein level [33], delivering the prerequisites for protein ubiquitinylation and degradation. Our data demonstrate the ubiquitous character of protein degradation pathways, which allows for the development of new technologies for directed proteolysis *via* the inducible expression of nanobody-SPOP fusions. Important regulatory functions may be blocked by the specific degradation of regulatory proteins in the nucleus.

## Supporting information

**S1 Fig. Pilot experiment to degrade EYFP-CENH3.** EYFP-CENH3 expressing *N. tabacum* plants were transiently transformed with either 35S::NSlmb-VHHGFP4 or 35S:: VHHGFP4-SPOP. Transient expression of 35S::NSlmb-VHHGFP4 leads to relocalisation of the fluorescence signal into the nucleoplasm. Addition of the VHHGFP-SPOP construct to EYFP-CENH3 expressing plants resulted in complete loss of fluorescence signals. (TIF)

**S2 Fig. PCR analysis of double-transformant plants.** Additional evidence of successful trans-formation was provided by PCR using genomic DNA and sequence-specific primers (S1 Table). (A) PCR amplified NSlmb sequence in three independently tested transgenic tobacco plant lines with NSlmb-VHHGFP/EYFP-CENH3 overexpression. The calculated length for

this construct is 1000 bp. The three lines tested contain the DNA sequence for SPOP expression. (B) Amplified SPOP in transgenic VHHGFP-SPOP/EYFP-CENH3 plant lines. The calculated molecular size of 1000 bp is obtained. The three lines tested contain the DNA sequence for SPOP expression. (C) Amplification of VHHGFP sequences in transgenic VHHGFP/EYFP-CENH3 plant lines. All three tested lines contain the DNA sequence for the nanobody. (D) Amplification of EYFP-CENH3 in control plant lines with EYFP-CENH3 overexpression. The calculated size of 700 bp was confirmed for all three tested plant lines. (E, F, G) Gel separation of the EYFP-CENH3 DNA constructs in NSlmb-VHHGFP4/EYFP-CENH3, VHHGFP4-SPOP/EYFP-CENH3 and VHHGFP4 EYFP-CENH3 transgenic plant lines. 10 individual plant lines were analysed for each transgenic combination.
(TIF)

**S3 Fig. Analysis of transgenic plant lines by confocal laser scanning microscopy.** Additional images. (A, B, C) NSlmb-VHHGFP4 EYFP-CENH3 transgenic plant leaf material. Images shown underline the tendency of EYFP-CEN3 to accumulate in the nucleoplasm when co-expressed with the NSlmb-VHHGFP4 construct. Different intensities of the fluorescence signal may result from varying copy numbers of the co-expressed E3-ligase NSlmb-VHHGFP4. (D, E, F) VHHGFP4-SPOP/ EYFP-CENH3 transgenic plant leaf material. Dramatically reduced or a complete lack of specific fluorescence signals was observed in these transgenic plants. Line 18 shows sporadically specific nuclear signals in stomata. (G, H, I) VHHGFP4/EYFP-CENH3 transgenic plant leaf material shows the typical centromeric fluorescence signal of EYFP-CENH3. (J) Control plant with overexpression of only EYFP-CENH3. Images of three independent plant lines are shown. Images were taken at 40x magnification.
(TIF)

**S4 Fig. Western blot analysis of double-transgenic tobacco plants.** (A) NSlmb-VHHGFP4/EYFP-CENH3 plant lines show a specific signal in the range of 38 kDa, that represents the overexpressed chimeric E3-ligase NSlmb-VHHGFP4. (B) Analysis of transgenic VHHGFP4-SPOP/EYFP-CENH3 overexpressing plant lines. The specific signal at 38 kDa indicates VHHGFP4-SPOP overexpression. (C) Analysed material from transgenic VHHGFP4 / EYFP-CENH3 plant lines. The signal within the range of the 15 kDa marker confirms the overexpression of the anti-GFP nanobody VHHGFP4. Visualisation of the target protein by C-terminal cmyc-tag *via* a specific monoclonal anti-cmyc antibody (4E10) and ECL-based detection. As internal Western blot control, 100 x ELP protein was used [35]. Faint additional bands below the 10 kDa marker are a result of chlorophyll autofluorescence in the gel running front. Results for three independent transgenic plant lines are shown. Computational quantification of signal strength of single bands was done by "LI-COR Image Studio" software (LI-COR Biosciences–GmbH, www.licor.com) designed for the analysis of Western blot images.
(TIF)

**S1 Table. Description and sequences of all primers used in this study.** The names describe the amplified DNA fragments. Sequences are presented in 5' to 3' direction.
(TIF)

## Acknowledgments

The authors thank Christine Helmold, Ingrid Pfort and Heike Kuhlmann for experimental help and Thomas Altmann for continous support.

## Author Contributions

**Conceptualization:** Udo Conrad.

**Funding acquisition:** Andreas Houben, Udo Conrad.

**Investigation:** Eberhard Sorge.

**Methodology:** Eberhard Sorge, Dmitri Demidov, Inna Lermontova.

**Project administration:** Udo Conrad.

**Supervision:** Andreas Houben, Udo Conrad.

**Visualization:** Eberhard Sorge, Dmitri Demidov, Inna Lermontova.

**Writing – original draft:** Eberhard Sorge.

**Writing – review & editing:** Dmitri Demidov, Inna Lermontova, Andreas Houben, Udo Conrad.

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
