## [Decision Letter · Decision Letter 0]

2 Dec 2020

PONE-D-20-35452

Engineered degradation of specific nuclear proteins via the 26S proteasome pathway in plants

PLOS ONE

Dear Dr. Sorge,

Thank you for submitting your manuscript to PLOS ONE. After careful consideration, we feel that it has merit but does not fully meet PLOS ONE’s publication criteria as it currently stands. Therefore, we invite you to submit a revised version of the manuscript that addresses the points raised during the review process.

I appreciate that the work presented here extends to nuclear proteins the use of chimeric E3 ligases to induce proteasomal degradation, and that the work has been carried out in N. benthamiana plants. However, both reviewers indicate that the impact of this particular technology would be higher if you could prove that it works under transient expression systems. Instead, I think that the most useful application would be to create conditional mutants using this technology, and testing how transgenic CEN3H-GFP can be degraded is only the starting point, but the endogenous CEN3H is still active in those plants. Even if not so impactful, your observations may help other researchers, so I am inclined towards acceptance of a future version that at least addresses the technical criticisms raised by the reviewers (western blots, replicates, etc).

We look forward to receiving your revised manuscript.

Kind regards,

Miguel A Blázquez

Academic Editor

PLOS ONE

Reviewers' comments:

Reviewer's Responses to Questions

**Comments to the Author**

1. Is the manuscript technically sound, and do the data support the conclusions?

Reviewer #1: Partly

Reviewer #2: Partly

2. Has the statistical analysis been performed appropriately and rigorously? 

Reviewer #1: N/A

Reviewer #2: Yes

3. Have the authors made all data underlying the findings in their manuscript fully available?

Reviewer #1: Yes

Reviewer #2: Yes

4. Is the manuscript presented in an intelligible fashion and written in standard English?

Reviewer #1: No

Reviewer #2: Yes

5. Review Comments to the Author

Reviewer #1: In this work, the authors create transgenic N. b. lines expressing constitutively EYFP-CenH3 (CenH3) alone, or in addition VHHYFP-NSImb (NSImb), VHHYFP-SPOP (SPOP) or VHHYFP. NSImb and SPOP are markers that due to their fusion to a YFP-binding nanobody address YFP or YFP:X fusion proteins for degradation by the proteasome. The authors want to show that this system works.

First they show that transcription of CenH3 is similar for most single transgenic and double transgenic plants. I agree that transcription is in the same magnitude of order (except for line 25), but there are still 6-fold differences in transcription levels.

Then the authors present confocal microscopy images (that are only shown in Fig S3) that indicate that CenH3 in CenH3 + NSImb double transgenic plants accumulates to similar protein levels as in CenH3 single transgenic plants. On the other hand, CenH3 accumulation in CenH3/SPOP plants is decreased. Thus NSImb does not seem to have an effect on CenH3 accumulation, and SPOP does have an effect.

Next the authors show western blots to detect NSImb and SPOP with anti-myc (again only shown in Fig S2). The blots are not interpretable because of the presence of strong bands that do not correspond to the expected weight. Please explain what is charged on the (+) lane. What is SNN (This applies also Figure 2)? Please explain what are the prominent low molecular weight bands revealed in S2A and S2B? They could be non-specific or degradation products of NSImb or SPOP.

Another series of western blots aims to detect CenH3. Unfortunately, the text and the legend did not allow me to understand what the blots are supposed to show. I also miss the lower part of the blots (around 20 kD) where the wild type CenH3 protein should be detected.

Finally, confocal images of transgenic plant epidermis are presented in Figure 4. It is not defined what is shown in 4A and 4B. Further, the authors do not indicate which transgenic lines were used for this experiment. This makes interpretation especially of images showing CenH3/NSImb transgenic leaves difficult, the further so because in Figure S3 (which essentially shows the same as Figure 4) different accumulation and intranuclear localization of CenH3 is observed in the different lines. The results are more clear for CenH3/SPOP lines, where CenH3 levels are reduced in all shown transgenic lines.

In my opinion, this work shows a preliminary analysis of using SPOP-YHHYFP (NSImb does not function in their system) to down-regulate protein levels of a YFP-tagged target protein. The system works for CenH3, if it can also be used efficiently for other proteins, remains unclear, because this was not tested.

If the authors intend to provide the community with an alternative system to reduce levels of a target protein, it might be more convenient to use agro-infiltration to introduce the target protein, instead of the time-consuming creation of a transgenic line.

Reviewer #2: In this work, Sorge and colleagues show an (engineered) system to degrade a specific nuclear protein (EYFP-CENH3) via the recruitment of the 26S proteasome pathway in Arabidopsis. Although the Ms is well written and the experiments well designed, in my opinion some extra experimental data should be included to strengthen this work. See my (two) major concerns and minor comments below.

1. In the abstract, authors claim that they overcame two of the limitations of using classical genetic approaches to knock out protein function in plants: time-consuming of generating homozygous transgenic lines, and the risk of non-viable loss-of-function phenotypes. I agree with the second one since a cenh3 null allele in Arabidopsis is embryo lethal. Regarding the first one the authors should show that this engineered degradation works in a transient infiltration (in Arabidopsis).

2. Is the Bradford assay compatible with a buffer containing 4%SDS?

Minor comments:

1- This paper should discuss or at least mention in the introduction: Wang et al (2020). Nat Plants 6:766-772.

2- Line 109: Nicotiana tabacum in italics.

3- The authors analyzed nuclear extracts and show H3 as loading control. However, I’d like also to see that there is no contamination of cytosolic proteins (for example, showing a ponceau staining). In the M&M section, authors indicate that they stained in Coomassie Blue but these results are not included.

4- How many biological replicates did the authors perform for each experiment? Please, specify this in all the figure legends or in the M&M section. It would be desirable to have at least 3 and to quantify the levels of proteins.

5- Line 198-202: “Transgenic plants showed comparable transcript levels, with 0.2 to 1.2 relative expression units”. It is quite obvious that there are differences in expression, at least from my view.

6- Fig S2: What does NNS mean?

6. PLOS authors have the option to publish the peer review history of their article (what does this mean?). If published, this will include your full peer review and any attached files.

Reviewer #1: **Yes: **Martin Drucker

Reviewer #2: No

---

## [Author Response · Author response to Decision Letter 0]

25 Jan 2021

Dear Miguel A Blázquez,

Many thanks for the professional handling of our manuscript and many thanks to both reviewers about their useful suggestions.

Below you will find our answers/changes in response to the suggested changes.

We colour coded the adjusted text of the manuscript. 

Our original blot/gel image data were posted at a public data repository with the following URL: http://dx.doi.org/10.5447/ipk/2021/1.

Yours sincerely,

Eberhard Sorge

Reviewers' comments:

Reviewer #1: 

1. Comment: First they show that transcription of CenH3 is similar for most single transgenic and double transgenic plants. I agree that transcription is in the same magnitude of order (except for line 25), but there are still 6-fold differences in transcription levels.

Response: Differences are discussed in line212-216: “But among the 11 tested lines, we observed one outlier with the genotype NSlmb-VHHGFP4/EYFP-CENH3 (line 25). The relative EYFP-CENH3 expression of this line was 12 times higher than that of the other lines; the higher expression was also visible as stronger fluorescence signals in the microscopy images (S3 Fig). Therefore, the presence of a degradation construct does not influence the transcript level of the target gene/protein. “ 

It is also now stated in the figure legend line 457-461: “The mRNA levels for EYFP-CENH3 varied between 0.2 and 1.2, except for the NSlmb-VHHGFP4 Line 25 with a value of 4. Fluorescence microscopy analysis confirmed that the high abundance of transcripts in line 4 corresponds to the high amount of the fusion protein (S3 Fig). Three biological and three technical replicates were analysed for each tested plant line.”

2. Comment: Then the authors present confocal microscopy images (that are only shown in Fig S3) that indicate that CenH3 in CenH3 + NSImb double transgenic plants accumulates to similar protein levels as in CenH3 single transgenic plants. On the other hand, CenH3 accumulation in CenH3/SPOP plants is decreased. Thus NSImb does not seem to have an effect on CenH3 accumulation, and SPOP does have an effect. 

Response: The main results of confocal microscopy are shown in figure 4. As the reviewer commented, the different accumulation patterns depend on the protein that is coexpressend with EYFP-CENH3. This is described in the results secion line 262-266: “The coexpression of NSlmb-VHHGFP4 and EYFP-CENH3 led to a strong accumulation of fluorescent signals in the nucleoplasm, additionally to reduced signals at the centromeres (Fig 4E and Fig 4F). The signal pattern changed from distinct fluorescent dots to diffuse signals, which likely represented non-centromeric chromatin. In contrast, VHHGFP4-SPOP/EYFP-CENH3 plants showed an overall reduction of fluorescence (Fig 4G and Fig 4H).”. Possible reasons for this are dicussed in line 292-297: “Neither the co-expression of an anti-GFP nanobody nor the co-expression of NSlmb-VHHGFP4 eliminated the nuclear EYFP signals. Western blot and epifluorescence data showed an even higher accumulation of the target protein after co-expression of VHHGFP4 or NSlmb-VHHGFP4. We propose that the nanobody we used for our studies stabilised the target protein, because GFP protein conformations can be manipulated with nanobodies even in living cells [11, 32].”

3. Comment: Next the authors show western blots to detect NSImb and SPOP with anti-myc (again only shown in Fig S2). The blots are not interpretable because of the presence of strong bands that do not correspond to the expected weight. Please explain what is charged on the (+) lane. What is SNN (This applies also Figure 2)? Please explain what are the prominent low molecular weight bands revealed in S2A and S2B? They could be non-specific or degradation products of NSImb or SPOP.

Response: This image has been reorganised (S4 Fig). The legend answers the raised questions: line: 527-528: “As internal Western blot control, 100 x ELP protein was used [35]. Additional bands below the 10 kDa marker are a result of chlorophyll autofluorescence in the gel running front.” 

4. Comment: Another series of western blots aims to detect CenH3. Unfortunately, the text and the legend did not allow me to understand what the blots are supposed to show. I also miss the lower part of the blots (around 20 kD) where the wild type CenH3 protein should be detected.

Response: The text (line 229-255) and the legend (line 465-478) were rewritten. The requested full size Western blots are shown as supplementary figures under DOI: http://dx.doi.org/10.5447/ipk/2021/1

The applied CENH3 antibody is specific for Arabidopsis-CENH3. Therefore, the detection of wild type Nicotiana CENH3 is not possible. 

5. Comment: Finally, confocal images of transgenic plant epidermis are presented in Figure 4. It is not defined what is shown in 4A and 4B. Further, the authors do not indicate which transgenic lines were used for this experiment. This makes interpretation especially of images showing CenH3/NSImb transgenic leaves difficult, the further so because in Figure S3 (which essentially shows the same as Figure 4) different accumulation and intranuclear localization of CenH3 is observed in the different lines. The results are more clear for CenH3/SPOP lines, where CenH3 levels are reduced in all shown transgenic lines.

Response: The content of figure 4A and 4B is now described in the legend line478: “(A and B) Transgenic plants accumulate CENH3-YFP within centromeric regions.”

And in the main text line 260-261: “Characteristic centromeric dot-like signals were found in EYFP-CENH3 plant lines (Fig 4A and Fig 4B).” Also the descriptions within the image have been extended for a better overview.

6. Comment: In my opinion, this work shows a preliminary analysis of using SPOP-YHHYFP (NSImb does not function in their system) to down-regulate protein levels of a YFP-tagged target protein. The system works for CenH3, if it can also be used efficiently for other proteins, remains unclear, because this was not tested.

Response: We agree and therefore narrowed down the title of the manuscript. New title: “Engineered degradation of EYFP-tagged CENH3 via the 26S proteasome pathway in plants”.

7. Comment: If the authors intend to provide the community with an alternative system to reduce levels of a target protein, it might be more convenient to use agro-infiltration to introduce the target protein, instead of the time-consuming creation of a transgenic line.

Response: We agree, an agroinfiltration experiment was performed as pilot experiment. These data are now included as as supplementary data. See Figure S1 and legend, and also main text line 185-191: “In early pilot experiments, we performed transient transformation to test whether the system is capable of degrading stably expressed EYFP-CENH3 by NSlmb-VHHGFP4 or VHHGFP4-SPOP. In transgenic N. tabacum lines with stable EYFP-CENH3 expression transiently transformed with a VHHGFP4-SPOP construct, a strong reduction of GFP signals was recognised (S1 Fig). Transient expression of NSlmb-VHHGFP4 in transgenic lines with stable EYFP-CENH3 did not lead to degradation but to relocalisation of the fluorescence signal.”

Reviewer #2:

1 Comment: 1. In the abstract, authors claim that they overcame two of the limitations of using classical genetic approaches to knock out protein function in plants: time-consuming of generating homozygous transgenic lines, and the risk of non-viable loss-of-function phenotypes. I agree with the second one since a cenh3 null allele in Arabidopsis is embryo lethal. Regarding the first one the authors should show that this engineered degradation works in a transient infiltration (in Arabidopsis).

Response: We agree, an agroinfiltration experiment was performed as pilot experiment in N. tabacum. These data are now included as as supplementary data. See Figure S1 and legend, and also main text line 185-191:: “In early pilot experiments, we performed transient transformation to test whether the system is capable of degrading stably expressed EYFP-CENH3 by NSlmb-VHHGFP4 or VHHGFP4-SPOP. In transgenic N. tabacum lines with stable EYFP-CENH3 expression transiently transformed with a VHHGFP4-SPOP construct, a strong reduction of GFP signals was recognised (S1 Fig). Transient expression of NSlmb-VHHGFP4 in lines with stable EYFP-CENH3 did not lead to degradation but to relocalisation of the fluorescence signal.”

2 Comment: Is the Bradford assay compatible with a buffer containing 4%SDS?

Response: Line 131-133: “For this purpose, 0.5 µl of each extract were mixed with 1 mL Bradford solution. Control values (0.5 µl 2xSDS sample buffer in 1 mL Bradford solution) were subtracted. “ The concentration of SDS in the Bradfort solution in the sample as well as in the control is then 0.02%. This allows relative concentration estimation at lower sensitivity of the assay, but the relatively high protein amount in the crude extracts could be determined without any problems. 

Minor comments:

Comment: 1- This paper should discuss or at least mention in the introduction: Wang et al (2020). Nat Plants 6:766-772.

Response: We added the suggested paper: line 49-50: “Here, a recent study showed an elegant way to reduce protein function in a cell-type-specific manner in plants, based on CRISPR/Cas9 and oestrogen- induceable promotor [3].”

Comment: 2- Line 109: Nicotiana tabacum in italics.

Response: corrected

Comment: 3- The authors analyzed nuclear extracts and show H3 as loading control. However, I’d like also to see that there is no contamination of cytosolic proteins (for example, showing a ponceau staining). In the M&M section, authors indicate that they stained in Coomassie Blue but these results are not included.

Response: We did not analyse the contamination of the nuclear extract with cytosolic proteins. We used Coomassie staining in combination with Bradford protein concentration measurement to apply the same amount of protein on the gel. But unfortunately we did not take a picture of the Coomossie stained gel. But since all samples of nuclear proteins were simultaneously isolated and the level of histone H3 based on our Western blot is comparable, the purity of nuclear fractions does not affect the quality of this experiment.

Comment: 4- How many biological replicates did the authors perform for each experiment? Please, specify this in all the figure legends or in the M&M section. It would be desirable to have at least 3 and to quantify the levels of proteins.

Response: A statement on the experimental material for figure 2 is added in line 460-461: “Three biological and three technical replicates were analysed for each tested plant line.” Likewise we added the requested information for figure 4 in line 484-485: “For all genotypes leaf material of 10 independent transgenic lines was analysed.” and for S3 Figure in line 517-518: “Images of three independent plant lines are shown. Images were taken at 40x magnification. “. We also added the in silico quanitifcation of protein levels, based on the Western blot, in Fig 3 and S4 Fig. The blots were performed once because there is no statistical relevance in repeating Western blots. It lies in the nature of the experiment to represent the state of many cells, when analysing crude plant extracts, 

Comment: 5- Line 198-202: “Transgenic plants showed comparable transcript levels, with 0.2 to 1.2 relative expression units”. It is quite obvious that there are differences in expression, at least from my view.

6- Fig S2: What does NNS mean?

Response: Differences are discussed in line:212-216: “But among the 11 tested lines, we observed one outlier with the genotype NSlmb-VHHGFP4/EYFP-CENH3 (line 25). The relative EYFP-CENH3 expression of this line was 12 times higher than that of the other lines; the higher expression was also visible as stronger fluorescence signals in the microscopy images (S3 Fig). Therefore, the presence of a degradation construct does not influence the transcript level of the target gene/protein.“ 

It is also now stated in the figure legend line 457-461: “The mRNA levels for EYFP-CENH3 varied between 0.2 and 1.2, except for the NSlmb-VHHGFP4 Line 25 with a value of 4. Fluorescence microscopy analysis confirmed that the high abundance of transcript in line 4 corresponds to the high amount of fusion protein (S3 Fig). Three biological and three technical replicates were analysed for each tested plant line.”

The abbreviation SNN stands for:N. tabacum L. cv. Samsun N. N.. 

We agree that his abbreviation was confusing and removed it from the manuscript. 

---

## [Editor Report · Decision Letter 1]

1 Feb 2021

Engineered degradation of EYFP-tagged CENH3 via the 26S proteasome pathway in plants

PONE-D-20-35452R1

Dear Dr. Sorge,

Thanks for satisfactorily addressing all the reviewers' queries. We’re pleased to inform you that your manuscript has been judged scientifically suitable for publication and will be formally accepted for publication once it meets all outstanding technical requirements.

Kind regards,

Miguel A Blázquez

Academic Editor

PLOS ONE
---

## [Editor Report · Acceptance letter]

3 Feb 2021

PONE-D-20-35452R1 

Engineered degradation of EYFP-tagged CENH3 via the 26S proteasome pathway in plants 

Dear Dr. Sorge:

I'm pleased to inform you that your manuscript has been deemed suitable for publication in PLOS ONE. Congratulations! Your manuscript is now with our production department. 

Kind regards, 

on behalf of

Dr. Miguel A Blázquez 

Academic Editor

PLOS ONE